# COVID-19 Vaccination Willingness among Chinese Adults under the Free Vaccination Policy

**DOI:** 10.3390/vaccines9030292

**Published:** 2021-03-21

**Authors:** Rugang Liu, Yuxun Zhang, Stephen Nicholas, Anli Leng, Elizabeth Maitland, Jian Wang

**Affiliations:** 1School of Health Policy & Management, Nanjing Medical University, Nanjing 211166, China; rugangliu@njmu.edu.cn (R.L.); yuxunzhang@njmu.edu.cn (Y.Z.); 2Center for Global Health, Nanjing Medical University, Nanjing 211166, China; 3Australian National Institute of Management and Commerce, Eveleigh, NSW 2015, Australia; stephen.nicholas@newcastle.edu.au; 4Research Institute for International Strategies, Guangdong University of Foreign Studies, Guangzhou 510420, China; 5School of Economics and School of Management, Tianjin Normal University, Tianjin 300074, China; 6Newcastle Business School, University of Newcastle, Newcastle, NSW 2308, Australia; 7School of Political Science and Public Administration, Institute of Governance, Shandong University, Qingdao 266237, China; lenganli@sdu.edu.cn; 8School of Management, University of Liverpool, Chatham Building, Chatham Street, Liverpool L69 7ZH, UK; e.maitland@liverpool.ac.uk; 9Dong Fureng Economic and Social Development School, Wuhan University, Wuhan 430072, China; 10Center for Health Economics and Management, School of Economics and Management, Wuhan University, Wuhan 430072, China

**Keywords:** COVID-19, vaccination willingness, free vaccination policy, determinant

## Abstract

(1) Background: China will provide free coronavirus disease 2019 (COVID-19) vaccinations for the entire population. This study analyzed the COVID-19 vaccination willingness rate (VWR) and its determinants under China’s free vaccination policy compared to a paid vaccine. (2) Methods: Data on 2377 respondents were collected through a nationwide questionnaire survey. Multivariate ordered logistic regression models were specified to explore the correlation between the VWR and its determinants. (3) Results: China’s free vaccination policy for COVID-19 increased the VWR from 73.62% to 82.25% of the respondents. Concerns about the safety and side-effects were the primary reason for participants’ unwillingness to be vaccinated against COVID-19. Age, medical insurance and vaccine safety were significant determinants of the COVID-19 VWR for both the paid and free vaccine. Income, occupation and vaccine effectiveness were significant determinants of the COVID-19 VWR for the free vaccine. (4) Conclusions: Free vaccinations increased the COVID-19 VWR significantly. People over the age of 58 and without medical insurance should be treated as the target intervention population for improving the COVID-19 VWR. Contrary to previous research, high-income groups and professional workers should be intervention targets to improve the COVID-19 VWR. Strengthening nationwide publicity and education on COVID-19 vaccine safety and effectiveness are recommended policies for decision-makers.

## 1. Introduction

At the end of January 2021, the World Health Organization reported over 102.1 million infected cases, and over 2.2 million deaths, globally from the coronavirus disease 2019 (COVID-19) [1]. Vaccinations will be the most effective and economic way to prevent COVID-19 and control its spread [2], and people’s vaccination willingness will decide whether they will receive the COVID-19 vaccine. During 2021, China’s COVID-19 prevention and control policy [3] will provide free COVID-19 vaccinations for high-risk cohorts and then the general population. The government’s free vaccination program, and the successful management of COVID-19, will depend on people’s vaccination willingness. High vaccination rates protect both the vaccinated and unvaccinated, create herd immunity and reduce the risk of virus mutations. This study analyzed the determinants affecting people’s COVID-19 vaccination willingness under China’s free vaccination policy compared to the paid vaccine. The survey was conducted in May–June 2020 before the announcement on 9 January 2021 of China’s free vaccination program.

Previous vaccination studies have shown that many factors are responsible for the COVID-19 vaccination willingness rate (VWR), such as socio-economic factors, awareness of the severity and susceptibility to the disease and trust in the vaccine [4,5,6,7,8]. From a survey in Japan, Yoda and Katsuyama found that males, older age people, rural residents and chronic disease sufferers displayed the highest willingness toward COVID-19 vaccination [4]. Marital status and trust in the health service system were found to be key determinants of COVID-19 vaccination willingness in Saudi Arabia [9]. Using an online survey of American adults, Reiter et al. found that participants were more likely to get vaccinated when they perceived a higher likelihood of getting a COVID-19 infection in the future, perceived a heightened severity of COVID-19 infection and perceived greater effectiveness in a COVID-19 vaccine; they were less likely to get vaccinated when they perceived higher potential vaccine harm [5]. Several studies have also been carried out on special populations, such as healthcare workers, long-term care staff and caregivers, which found that concerns about vaccine safety, side effects and effectiveness were the primary reasons for vaccine hesitancy [7,10,11,12,13]. A Chinese national online survey found that participants that perceived the benefits and were unconcerned about the efficacy of COVID-19 vaccines had the highest intention to vaccinate [6]. Their willingness-to-pay (WTP) for COVID-19 vaccines was influenced by social and demographic factors, such as occupation and region. A number of discrete choice experiments revealed that a strong preference for the COVID-19 vaccine depended on its effectiveness, side-effects, protection duration and number of injections [14,15,16,17]. However, little is currently known about the effect of free vaccinations on the COVID-19 VWR in China. To address this lacuna, we surveyed participants’ (un)willingness to vaccinate when the vaccine was free; identified the reasons participants would get vaccinated or not; and recommend measures to improve the COVID-19 VWR.

## 2. Materials and Methods

### 2.1. Data Source and Sample

A questionnaire was designed to collect COVID-19 vaccination willingness and other variables. China’s 27 provinces were divided into three regions: eastern, central and western. The provinces in each region were stratified into low, medium and high economic levels according to their 2019 gross domestic product (GDP). Randomly, one province was chosen from each economic level in each region, yielding nine provinces. Next, according to their 2019 GDP rank, all the cities in each selected province were divided into low, medium and high economic levels. One city was randomly chosen from each GDP level, with 27 cities selected from the 9 provinces. One hundred participants were interviewed face-to-face, or by online video interviews in cities where participants were required to home quarantine, in each city, with equal numbers of men and women and three urban residents for every two rural residents, which reflected the nationwide urban–rural breakdown. All investigators recruited in the 27 cities received standardized training before the formal investigations. During 30 May to 10 June 2020, face-to-face interviews were conducted by interviewers. All participants were informed about the purpose of the survey and gave informed consent. We collected data on 2700 adults over the age of 18 years old, which yielded a sample of 2377 respondents after deleting cases with missing data, with a response rate of 88.04%.

### 2.2. Definition and Measurement of Dependent Variables

The categorical dependent variable, COVID-19 vaccination willingness, was assessed by two questions: “Would you pay for the COVID-19 vaccination?” and “Would you get the COVID-19 vaccination if the vaccine were free?” There were three answers (“no” (0), “it depends” (1) and “yes” (2)) for each question, representing COVID-19 vaccination willingness being low, medium and high.

### 2.3. Definition and Measurement of Independent Variables

As shown in Table 1, the independent variables comprised sex (male–female), age groups, three average monthly income groups (low (<RMB4000), medium (≥RMB4000–<RMB8000) and high (≥RMB8000)), education level (below high school, and high school and above), occupation, medical insurance (yes/no), urban–rural residence, self-rated health, residence in east–west–central region, and awareness of COVID-19 vaccine effectiveness, safety and risk of infection. Three urban residents were interviewed for every two rural participants. Occupations were categorized into professionals (including physicians, teachers and civil servants), farmers, students, self-employed, unemployed, migrant workers and other. Self-rated health was categorized into, “bad”, “medium” and “good”, based on the question: “How is your health status compared to your peers?” Participant’s awareness of COVID-19 vaccine effectiveness and safety was measured by the questions: “Do you believe that the COVID-19 vaccine is effective?”/“Do you believe that the COVID-19 vaccine is safe?”, and coded into a three-item Linkert scale (“don’t agree—low effectiveness/safety”, “neutral attitude--medium effectiveness/safety” and “agree—high effectiveness/safety”). The respondents’ risk of infection was measured by asking participants whether they would be infected by COVID-19 in the future according to “low–neutral–high” measure of risk.

### 2.4. Statistical Analyses

All data were double-entered using EpiData 3.1 and checked for consistency. Statistical analyses were performed using STATA 12.0. The Pearson chi-square test was used to compare the differences in VWRs among different subgroups and the VWR differences between the paid and free vaccine. Multivariate ordered logistic regression models and odds ratio (OR) were used to assess the associations between each independent variable and the VWRs of the paid and free COVID-19 vaccines.

## 3. Results

### 3.1. Characteristics of Respondents

Table 1 shows the characteristics of 2377 survey respondents, with the male (49%) and female (51%) sex ratio broadly even and the urban (62%)–rural (38%) split close to the 3:2 national urban–rural ratio; the median age was 35; the median monthly income was RMB5000 and the three low-, medium- and high-income groups were roughly equal. In terms of education, 38.07% respondents had a high school and above education level. Professionals (29.79%) accounted for the highest occupational group, followed by students (26.88%), migrant workers (12.16%) and farmers (11.70%). Only 3.07% reported their self-assessed health as “bad” and 96% had one or more type of medical insurance. There was a broadly equal number of respondents from the eastern (31.47%), central (28.86%) and western (39.67%) regions. Respondents mainly believed that the COVID-19 vaccine was effective (86.62%); 82.41% believed that the vaccine was safe and 53.43% thought they were at medium or high risk of COVID-19 infection.

### 3.2. COVID-19 Vaccination Willingness Rate

Figure 1 shows that the COVID-19 VWR increased from 73.62% to 82.25% when vaccination changed from paid to free, and the uncertain group fell from 20.7% to 14.05% (Chi^2^ = 827.89, *p* < 0.001). For the paid vaccine, only 5.7% were unwilling to vaccinate: first, because of concerns about the safety and side-effects of the COVID-19 vaccine (32.99%); second, because there was no perceived need for vaccination because the COVID-19 outbreak had been controlled in China (23.71%); and third, because of the expense of the vaccine (15.46%). For the free vaccine, only 3.7% were unwilling to vaccinate, with 51.32% worried about the safety and side-effects of COVID-19 vaccines; 13.16% did not believe that the vaccine was effective and 9.21% believed that they would not be infected in the future and that it was unnecessary to vaccinate against COVID-19.

### 3.3. Results of Chi-Square Test

Table 2 shows that the COVID-19 VWR for the free vaccine was higher than that of the paid vaccine in all subgroups (*p* < 0.05), except for the low vaccine effectiveness group, the VWR of which was 47.06% for the paid vaccine and 41.18% for the free vaccine (*p* = 0.005). For both the paid and free vaccine, the COVID-19 VWRs of the below high school education group was higher than the high school and above group (*p* < 0.05); farmers retained the highest VWR among all occupations, with a 76.62% VWR for the paid vaccine and 88.85% VWR for the free vaccine (*p* = 0.001); the VWR of respondents who had medical insurance was higher than that of non-insured respondents (*p* < 0.05); and the VWR increased with participants’ increased awareness of vaccine effectiveness and safety (*p* < 0.001). The VWR of the 28–37 age group was the highest, and the VWR of the over-58 age group was the lowest for the paid vaccine group, with a VWR of 69.49% (*p* < 0.001), but it rose to 83.9% for the free vaccine (*p* = 0.001). There were no differences among income groups for the paid vaccine, but the VWR decreased with income when the vaccine was free (*p* = 0.004), with the highest VWR for the low-income group (86.2%). The central region had the highest VWR (83.53%), followed by the western area (82.61%) and the eastern area (80.61%), for the free vaccine (*p* = 0.034). The participants with the highest awareness of a COVID-19 infection risk were most willing to get vaccinated, with a VWR of 84.33% for the free vaccine (*p* = 0.013).

### 3.4. Results of Multiple Ordered Logistic Regressions

Table 3 shows the results of two multivariate ordered logistic regression models specified to analyze the relationship between the VWR and the independent variables for the paid and free vaccine. Sex, education level, urban–rural residence, self-rated health, region and awareness of infection risk had no influence on VWR for both paid and free vaccines. The COVID-19 VWR of respondents aged above 58 was significantly lower than the 18–27 age group (OR = 0.581, *p* = 0.019 for the paid vaccine; OR = 0.456, *p* = 0.006 for the free vaccine) and respondents without medical insurance had a lower COVID-19 VWR than respondents who had medical insurance (OR = 0.496, *p* = 0.002 for the paid vaccine; OR = 0.513, *p* = 0.011 for the free vaccine). The COVID-19 VWR increased with vaccine safety for both the paid (OR = 2.061, *p* = 0.037 (medium group); OR = 4.692, *p* < 0.001 (high group)) and free (OR = 2.071, *p* = 0.046 (medium group); OR = 6.641, *p* < 0.001 (high group)) vaccine. There was no difference in VWR among different income groups for the paid vaccine, but the COVID-19 VWR for the low-income group was higher than for high-income group for the free vaccine (OR = 1.536, *p* = 0.009). There was no statistical difference between the VWRs of famers and professionals for the paid vaccine, while the farmers had a higher VWR than professionals when the vaccine was free (OR = 2.016, *p* = 0.008). Self-employed respondents had a lower VWR than professional participants for the paid vaccine (OR = 0.695, *p* = 0.046), but there was no difference between the self-employed and professionals in the free vaccine model in Table 3. For the paid vaccine, there was no difference in the COVID-19 VWRs among respondents with different awareness levels of vaccine effectiveness; however, the VWR of people with a high awareness level of vaccine effectiveness was higher than the low awareness group (OR = 5.49, *p* = 0.002).

## 4. Discussion

The COVID-19 VWR and its influencing factors among Chinese adults were analyzed for a paid and a free vaccine. The outcomes can be used to guide projections of future COVID-19 vaccine uptake and shape vaccination policy. The COVID-19 VWR was 82.25% for the free vaccine, significantly higher than 73.62% for the paid vaccine, increasing the COVID-19 VWR by 11.72%. For the paid vaccine, the VWR in this study was higher than in similar studies in Japan (65.7%) [4], France (62%), Germany (70%) [18] and the United States (69%) [5], but roughly the same as the Netherlands (73%), Italy (74%) and Portugal (75%); the VWR for the free vaccine was similar to Denmark (80%) [18] and Australia (85.8%) [19].

Age, medical insurance and vaccine safety were significant influencing factors in both the paid and free vaccine regression models; sex, education level, urban–rural residence, self-rated health, region and awareness of infection risk were not significant factors in VWR. Although free vaccinations raised COVID-19 VWR from 69.49% to 83.9% for respondents aged over 58 years old, their VWR remained lower than the 18–27 age group for both paid (OR = 0.581) and free vaccinations (OR = 0.456). This finding was counter to previous studies in Japan and the United States, which showed that the elderly had a higher COVID-19 vaccination willingness [4,20]. The COVID-19 VWR of people who had no medical insurance was only half that of those who had medical insurance (OR = 0.496 for the paid vaccine, OR = 0.513 for the free vaccine) even through the COVID-19 VWR of people without medical insurance increased from 56.82% to 71.59% when the vaccine was free. This finding is similar to past research, which found the COVID-19 vaccine acceptance of people with private or public health insurance was higher than people without health insurance [5]. One recommendation from our study is that special measures should be taken among people aged over 58 years old and without medical insurance to enhance their VWR.

Vaccine safety had the greatest impact on COVID-19 VWR, with the biggest OR = 6.63 for the high vaccine safety group in the free vaccine model. The COVID-19 VWR increased when respondents’ awareness of vaccine safety increased, consistent with a U.S. study that found that perceptions of the potential harm of a COVID-19 vaccine decreased people’s willingness to get vaccinated. However, the U.S. study also found that perceiving a risk of getting a COVID-19 infection in the future increased the VWR, which was different from the result in our study [5]. A study in Kuwait also showed that the respondents were less willing to accept COVID-19 vaccination when they viewed vaccines in general to have health-related risks [21]. We suggest that improving people’s awareness of vaccine safety will help to increase the COVID-19 VWR.

Income had no influence on COVID-19 VWR in the paid vaccine model, but the low-income group had a higher COVID-19 VWR than the high-income group in the free vaccine model (86.2% vs. 78.97%, OR = 1.536), which was different from previous research [5,6,22,23,24,25]. The VWR growth rate of free vaccinations for the low-income group was 17.37% (from 73.44% to 86.2%), but that for the high-income group was only 6.64% (from 74.05% to 78.97%). Free vaccinations had the greatest effect on the COVID-19 VWR of low-income people among all income groups. Surprisingly, one recommendation from our study is that new interventions should target the high-income group because their COVID-19 VWR was lower than the low-income group when vaccinations were free.

The COVID-19 VWR of self-employed people was lower than professionals for the paid vaccine. Famers’ COVID-19 VWR increased from 76.62% to 88.85% with a growth rate of 15.96%, and that of self-employed people increased from 68.78% to 81.45% when the vaccine was free. The COVID-19 VWR of professionals only increased from 74.15% to 78.95%, with a growth rate of 6.47%, and their VWR was also lower than famers’ VWR for the free vaccine. While the self-employed should be a key intervention target for the paid vaccine, professionals are a key intervention group for free vaccinations.

For the paid vaccine, people’s COVID-19 VWR was only affected by vaccine safety, without any influence from vaccine effectiveness in the regression analysis. For the free vaccine, people cared not only about vaccine safety, but also about vaccine effectiveness. These results are similar to previous studies that showed that the perceived effectiveness of a COVID-19 vaccine improved the VWR [5,6,14,15,26]. Worrying about the safety and side-effects was also a main reason for people’s unwillingness to be vaccinated for both the paid vaccine (32.99%) and free vaccine (51.32%). Suspecting the effectiveness of vaccines was the second reason for unwillingness to take up the free vaccine (13.16%) in this study. For both the paid and free vaccines, information and education campaigns should be launched to reassure the public that listed COVID-19 vaccines are safe and effective and thus enhance the public’s trust in the COVID-19 vaccines [18,27,28,29,30]. Since urban–rural residence and regional location were not significant determinants of the VWR, information and education campaigns can be national.

### Strengths and Limitations

This study has several strengths. First, this was the first study to assess the effect of a COVID-19 free vaccination policy on the COVID-19 VWR among Chinese adults. Second, the dataset came from a nationwide questionnaire survey, covering both rural and urban areas, and three (eastern, central and western) regions in China. Third, face-to-face or online video interviews were conducted to collect the data in contrast to most past research, which relied on online questionnaire surveys.

There were two major limitations. First, we used only one question to evaluate respondents’ COVID-19 vaccination willingness; a more complex vaccination willingness scale should be developed in further studies. Second, the sample was limited to only 100 respondents in each city across three regions in China. 

## 5. Conclusions

Free vaccinations increased COVID-19 VWR by 11.72%, increasing the willingness to vaccinate from 73.62% for the paid vaccine to 82.25% for the free vaccine. Given the negative health and economic impact of the COVID-19 virus, increased vaccination rates promotes herd immunity and China’s return to pre-COVDI-19 normality. For both the paid and free vaccine, vaccine safety was the most significant concern among the study participants. The COVID-19 VWR for the free vaccine for respondents aged over 58 years old and without medical insurance was lower than the reference groups. Our results suggest that intervention measures should be directed towards people aged over 58 years old and without medical insurance to improve their COVID-19 VWR. For the free vaccine, the COVID-19 VWR of the high-income group was lower than the low-income group and that of professionals lower than farmers. Surprisingly, our results suggest that high-income groups and professionals should be special COVID-19 VWR intervention targets for the free vaccine. Strengthening COVID-19 vaccine knowledge through public information and education campaigns on vaccine safety would improve VWR, whether the vaccine is free or not. Vaccine effectiveness should be included in a nationwide COVID-19 education campaign, especially for the free vaccine.

## Figures and Tables

**Figure 1 vaccines-09-00292-f001:**
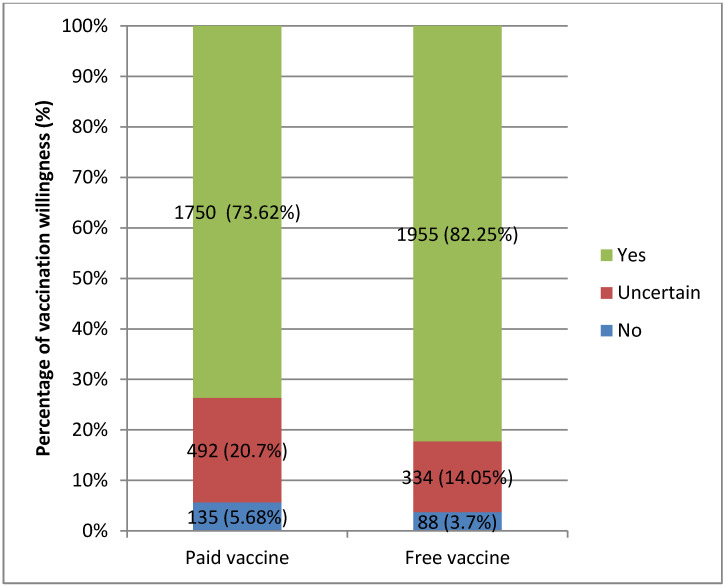
Comparison of COVID-19 vaccination willingness rates (VWRs) between paid and free vaccine.

**Table 1 vaccines-09-00292-t001:** Characteristic of respondents.

Variables		N	%
Sex	Male	1154	48.55
	Female	1223	51.45
Age	18–27	915	38.49
	28–37	340	14.30
	38–47	460	19.35
	48–57	426	17.92
	58+	236	9.93
Income	High (≥RMB8000)	813	34.20
	Medium (≥RMB4000–<RMB8000)	796	33.49
	Low (<RMB4000)	768	32.31
Education level	Below high school	1472	61.93
	High school and above	905	38.07
Occupation	Professional	708	29.79
	Farmer	278	11.70
	Migrant worker	289	12.16
	Self-employed	221	9.30
	Unemployed	103	4.33
	Student	639	26.88
	Retired	86	3.62
	Other	53	2.23
Medical insurance	Yes	2289	96.30
	No	88	3.70
Residence	Urban	1462	61.51
	Rural	915	38.49
Self-rated health	Bad	73	3.07
	Medium	564	23.73
	Good	1740	73.20
Region	Eastern	748	31.47
	Central	686	28.86
	Western	943	39.67
Vaccine effectiveness	Low	17	0.72
	Medium	301	12.66
	High	2059	86.62
Vaccine safety	Low	40	1.68
	Medium	378	15.90
	High	1959	82.41
Risk of infection	Low	1107	46.57
	Medium	670	28.19
	High	600	25.24

**Table 2 vaccines-09-00292-t002:** Comparison of COVID-19 vaccination willingness for paid and free vaccination with independent variables.

Variables		VWR of Paid Vaccine				VWR of Free Vaccine				Comparison between Paid and Free Vaccination	
		N	%	χ^2^	*p*	N	%	χ^2^	*p*	χ^2^	*p*
Sex	Male	862	74.70	1.50	0.471	953	82.58	0.72	0.698	498.37	<0.001
	Female	888	72.61			1002	81.93			346.51	<0.001
Age	18–27	670	73.22	63.75	<0.001	755	82.51	25.12	0.001	244.42	<0.001
	28–37	266	78.24			291	85.59			93.89	<0.001
	38–47	321	69.78			362	78.70			185.32	<0.001
	48–57	329	77.23			349	81.92			208.45	<0.001
	58+	164	69.49			198	83.90			66.22	<0.001
Income	High	602	74.05	2.95	0.565	642	78.97	15.35	0.004	339.79	<0.001
	Medium	584	73.37			651	81.78			271.24	<0.001
	Low	564	73.44			662	86.20			235.30	<0.001
Education level	Below high school	1093	74.25	22.71	<0.001	1228	83.42	7.68	0.021	536.48	<0.001
	High school and above	657	72.60			727	80.33			280.16	<0.001
Occupation	Professional	525	74.15	36.93	0.001	559	78.95	35.57	0.001	290.55	<0.001
	Farmer	213	76.62			247	88.85			99.27	<0.001
	Migrant worker	220	76.12			239	82.70			104.51	<0.001
	Self-employed	152	68.78			180	81.45			78.91	<0.001
	Unemployed	71	68.93			89	86.41			29.02	<0.001
	Student	468	73.24			525	82.16			184.11	<0.001
	Retired	63	73.26			73	84.88			43.66	<0.001
	Other	38	71.70			43	81.13			13.47	0.009
Medical insurance	Yes	1700	74.27	13.66	0.001	1892	82.66	8.39	0.015	786.92	<0.001
	No	50	56.82			63	71.59			36.37	<0.001
Residence	Urban	1080	73.87	0.84	0.657	1197	81.87	0.39	0.821	582.10	<0.001
	Rural	670	73.22			758	82.84			258.51	<0.001
Self-rated health	Bad	50	68.49	7.79	0.100	60	82.19	1.17	0.883	11.14	0.025
	Medium	400	70.92			466	82.62			161.68	<0.001
	Good	1300	74.71			1429	82.13			680.67	<0.001
Region	Eastern	546	72.99	6.80	0.147	603	80.61	10.40	0.034	265.17	<0.001
	Central	505	73.62			573	83.53			386.46	<0.001
	Western	699	74.13			779	82.61			188.60	<0.001
Vaccine effectiveness	Low	8	47.06	124.14	<0.001	7	41.18	245.71	<0.001	14.85	0.005
	Medium	147	48.84			162	53.82			138.87	<0.001
	High	1595	77.46			1786	86.74			529.37	<0.001
Vaccine safety	Low	18	45.00	167.88	<0.001	21	52.50	279.62	<0.001	25.40	<0.001
	Medium	194	51.32			214	56.61			199.46	<0.001
	High	1538	78.51			1720	87.80			421.77	<0.001
Risk of infection	Low	802	72.45	9.22	0.056	914	82.57	12.69	0.013	394.98	<0.001
	Medium	485	72.39			535	79.85			250.14	<0.001
	High	463	77.17			506	84.33			206.12	<0.001

**Table 3 vaccines-09-00292-t003:** Results of multiple ordered logistic regressions.

Variables		Paid Vaccine				Free Vaccine			
		β	S.E.	*p*	OR (95% CI)	β	S.E.	*p*	OR (95% CI)
Sex	Male	(Reference group)				(Reference group)			
	Female	−0.09	0.10	0.335	0.910 (0.752, 1.102)	−0.11	0.12	0.353	0.898 (0.715, 1.127)
Age	18–27	(Reference group)				(Reference group)			
	28–37	0.28	0.19	0.146	1.325 (0.907, 1.937)	0.15	0.24	0.521	1.164 (0.732, 1.849)
	38–47	−0.14	0.18	0.426	0.869 (0.614, 1.229)	−0.32	0.21	0.130	0.724 (0.476, 1.100)
	48–57	0.07	0.19	0.706	1.075 (0.739, 1.562)	−0.37	0.23	0.101	0.691 (0.445, 1.075)
	58+	−0.54	0.23	0.019	0.581 (0.370, 0.913)	−0.79	0.29	0.006	0.456 (0.260, 0.799)
Income	High	(Reference group)				(Reference group)			
	Medium	−0.06	0.12	0.645	0.945 (0.743, 1.202)	0.17	0.14	0.223	1.187 (0.901, 1.565)
	Low	−0.06	0.14	0.661	0.942 (0.723, 1.229)	0.43	0.16	0.009	1.536 (1.113, 2.118)
Education level	Below high school	(Reference group)				(Reference group)			
	High school and above	−0.16	0.12	0.197	0.855 (0.673, 1.085)	−0.21	0.14	0.146	0.814 (0.617, 1.074)
Occupation	Professional	(Reference group)				(Reference group)			
	Farmer	0.22	0.21	0.283	1.247 (0.833, 1.865)	0.70	0.26	0.008	2.016 (1.205, 3.373)
	Migrant worker	0.08	0.18	0.649	1.086 (0.761, 1.550)	0.10	0.21	0.616	1.111 (0.737, 1.673)
	Self-employed	−0.36	0.18	0.046	0.695 (0.486, 0.993)	0.03	0.22	0.887	1.032 (0.672, 1.584)
	Unemployed	−0.07	0.26	0.793	0.933 (0.556, 1.566)	0.62	0.35	0.077	1.859 (0.936, 3.693)
	Student	−0.04	0.18	0.800	0.956 (0.676, 1.353)	−0.11	0.21	0.610	0.897 (0.591, 1.361)
	Retired	0.13	0.29	0.663	1.137 (0.638, 2.024)	0.58	0.37	0.111	1.789 (0.875, 3.659)
	Other	−0.16	0.33	0.637	0.854 (0.445, 1.641)	0.05	0.40	0.901	1.050 (0.483, 2.284)
Medical insurance	Yes	(Reference group)				(Reference group)			
	No	−0.70	0.22	0.002	0.496 (0.321, 0.766)	−0.67	0.26	0.011	0.513 (0.306, 0.858)
Residence	Urban	(Reference group)				(Reference group)			
	Rural	−0.05	0.11	0.666	0.955 (0.773, 1.178)	−0.09	0.13	0.477	0.913 (0.711, 1.173)
Self-rated health	Bad	(Reference group)				(Reference group)			
	Medium	0.23	0.28	0.413	1.262 (0.723, 2.204)	0.34	0.36	0.333	1.411 (0.703, 2.834)
	Good	0.34	0.28	0.230	1.399 (0.808, 2.422)	0.27	0.35	0.443	1.306 (0.660, 2.584)
Region	Eastern	(Reference group)				(Reference group)			
	Central	0.03	0.13	0.823	1.029 (0.800, 1.325)	0.08	0.15	0.593	1.085 (0.804, 1.464)
	Western	0.08	0.12	0.485	1.088 (0.859, 1.378)	−0.03	0.14	0.850	0.974 (0.738, 1.284)
Vaccine effectiveness	Low	(Reference group)				(Reference group)			
	Medium	−0.05	0.52	0.920	0.950 (0.344, 2.622)	0.80	0.54	0.138	2.229 (0.773, 6.430)
	High	0.66	0.52	0.202	1.931 (0.703, 5.303)	1.70	0.54	0.002	5.490 (1.910, 15.785)
Vaccine safety	Low	(Reference group)				(Reference group)			
	Medium	0.72	0.35	0.037	2.061 (1.044, 4.068)	0.73	0.36	0.046	2.071 (1.013, 4.234)
	High	1.55	0.34	<0.001	4.692 (2.411, 9.131)	1.89	0.36	<0.001	6.641 (3.279, 13.450)
Risk of infection	Low	(Reference group)				(Reference group)			
	Medium	0.17	0.11	0.146	1.181 (0.944, 1.478)	0.06	0.14	0.657	1.062 (0.814, 1.386)
	High	0.20	0.12	0.101	1.221 (0.962, 1.550)	0.03	0.15	0.819	1.034 (0.778, 1.374)

## Data Availability

The data presented in this study are available on request from the corresponding author. The data are not publicly available due to multi-cooperation with Wuhan University, Shandong University and Nanjing Medical University. The corresponding author will facilitate a discussion with these three universities for data access on a reasonable request.

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
