# Peer review of "COVID-19 Vaccination Willingness among Chinese Adults under the Free Vaccination Policy"

_vaccines, 2021, doi:10.3390/vaccines9030292_

Round 1
Reviewer 1 Report
There are several similar studies from high income countries in Europe, North America and Japan. It is therefore of interest to read a similar paper from another part of the world, China, the presumed country of the origin of covid 19. The authors write in the introduction that one aim of the study is measurements should be undertaken to increase the willingness to get vaccinated. I think they could stress the importance of high vaccination rates. As long as the virus exists in the human population, there is a risk that it mutates and that some mutants can be vaccine resistant. I do not see anywhere in the manuscript how high the response rate was, i.e. how many of those contacted actually were interviewed fac-to face of with the written questionnaire.
Two details:
Line 117. I have never seen the word ”in-putted”. I think ”entered woud be better.
Line 125: they write ”male (51%) 125 and female (49%)”. This does not agree with Table 1, according to which the were 51 % women and 49 % men.
Author Response
Dear Professor Zorana Petrovic and reviewers,
Thank you for reviewing our manuscript (ID: vaccines-1127362) and giving us the opportunity to respond to the reviewers’ concerns. We have made changes to address each of the concerns of the reviewers. Our responses, point-by-point, to the comments are described below.
Editor's comments
- It is important to confirm your author list and the corresponding affiliations. The authors' names and email addresses in the manuscript should be consistent with those in the system. Please, also check your data, contents, acknowledgments, funding, etc. If the manuscript is accepted for publication, we would not accept any further modification of these issues.
- The number of references should be more than 30, please revise this.
- Please revise the word "War" in the manuscript.
Authors’ response:
Thank you for the valuable comments. The author list and the corresponding affiliations have been checked. We also added the number of references to 30. We have checked the manuscript to ensure that it does not include the word “War”.
Reviewer's comments:
Reviewer#1:
Comments and Suggestions for Authors
There are several similar studies from high income countries in Europe, North America and Japan. It is therefore of interest to read a similar paper from another part of the world, China, the presumed country of the origin of covid 19. The authors write in the introduction that one aim of the study is measurements should be undertaken to increase the willingness to get vaccinated. I think they could stress the importance of high vaccination rates. As long as the virus exists in the human population, there is a risk that it mutates and that some mutants can be vaccine resistant. I do not see anywhere in the manuscript how high the response rate was, i.e. how many of those contacted actually were interviewed fac-to face of with the written questionnaire.
Authors’ response:
On page 3, lines 101, we indicated the response rate. On page 2, lines 96-98, we explained that online video interviews were used in cities where participants were required to home quarantine. On page 2, lines 50-51, we addressed in a sentence the issue of high vaccination rates, herd immunity and virus mutation.
Two details:
Line 117. I have never seen the word ”in-putted”. I think ”entered would be better.
Authors’ response:
Thanks for your comments. “in-putted” was replaced by “entered”. (Line 128).
Line 125: they write ”male (51%) 125 and female (49%)”. This does not agree with Table 1, according to which the were 51 % women and 49 % men.
Authors’ response:
We have corrected this mistake. (Line 136-137)
Reviewer#2:
Comments and Suggestions for Authors
The manuscript by Rugang et al, on “COVID-19 Vaccination Willingness among Chinese Adults under the Free Vaccination Policy”. The study is well designed and the experiments were sequentially performed well. The data strongly support the conclusion made and the methods used are well documented.
Authors’ response:
Thanks for your positive comments.
Reviewer#3:
Comments and Suggestions for Authors
Liu et al.'s article presents a survey of COVID-19 vaccine willingness rate (VWR) to examine participants' (un)willingness to vaccinate when the vaccine is free or paid; identified the reasons for participants to get vaccinated/not vaccinated; and recommended measures to improve the COVID-19 VWR. They found that China's free vaccination policy of COVID-19 increased VWR significantly from 73.62% to 82.25% of the respondents. Concerns about the safety and side-effects were the primary reason for participants' unwillingness to get COVID-19 vaccination. Age, medical insurance, and vaccine safety were significant determinants of COVID-19 VWR for both the paid and free vaccine. Income, occupation, and vaccine effectiveness were significant determinants of COVID-19 VWR for the free vaccine. Overall, the authors suggested targeting people over 58 years without medical insurance to improve COVID-19 VWR as well as the high-income groups for free COVID vaccine, and nationwide measures to educate people and increase awareness were needed. Overall, the article is well written and presents substantial data, the results are well explained, and a good discussion of the presented data and its comparison with previous studies from other parts of the world is provided. Additionally, the author also points out the strengths and weaknesses of the article and future steps to improve. Thus, I think the manuscript will be a valuable addition to COVID-19 vaccination research to understand VWR and measures to improve. I have no hesitation in accepting the manuscript for publication, once the below minor comments are attended to-
- Section 2.1 needs to be improved and explained a bit more in detail. The random selection of provinces and cities is a bit vague, and thus, needs to be explained a bit more – how was it picked? Also, it says 2700 adults were interviewed that yielded 2377 respondents, the change in the number of participants is not clear.
Authors’ response:
We have explained the random selection of provinces and cities in section “2.1. Data Source and Sample” (Lines 85-91) and we have explained the change in the number of participants. (Line 101)
- When was this study conducted? For how long-duration of the study, and the order of the participants- which group/province, city, etc., were interviewed first?- All of these needs to be included.
Authors’ response:
We have indicated the dates of the survey (lines 53-55, 96) and added additional descriptions about conducting the survey in section “2.1. Data Source and Sample”. (Lines 95-98)
- When was China's free vaccination policy introduced? And were these interviews conducted after the policy was issued? Or under the assumption that such policy might be issued?
Authors’ response:
These interviews were conducted under the assumption that such policy might be issued. We have made this clear in lines 53-55.
- Why was the urban to rural ratio kept around 3:2? This should be explained.
Authors’ response:
We have explained this ratio in section “2.1. Data Source and Sample”. (Line 94-95)
- Figure 1: add a description to the y-axis
Authors’ response:
The description to the y-axis was added in Figure 1.
- Page 1, line 30, change "unwilling" to "unwillingness"
Authors’ response:
"unwilling" was changed to "unwillingness".(Line 30)
- Page 2, line 74-75- change –"identified the reasons participants vaccinated/not vaccinated" to "identified the reasons participants will get vaccinated/not vaccinated"
Authors’ response:
"identified the reasons participants vaccinated/not vaccinated" was changed into "identified the reasons participants will get vaccinated/not vaccinated". (Line 79)
- Page 9, line 270- change "draw" to "drawn"
Authors’ response:
"draw" was changed to "drawn". (Line 285)
Reviewer 2 Report
The manuscript by Rugang et al, on “COVID-19 Vaccination Willingness among Chinese Adults under the Free Vaccination Policy”. The study is well designed and the experiments were sequentially performed well. The data strongly support the conclusion made and the methods used are well documented.
Author Response

(The authors gave the same response as above.)

Reviewer 3 Report
Liu et al.'s article presents a survey of COVID-19 vaccine willingness rate (VWR) to examine participants' (un)willingness to vaccinate when the vaccine is free or paid; identified the reasons for participants to get vaccinated/not vaccinated; and recommended measures to improve the COVID-19 VWR. They found that China's free vaccination policy of COVID-19 increased VWR significantly from 73.62% to 82.25% of the respondents. Concerns about the safety and side-effects were the primary reason for participants' unwillingness to get COVID-19 vaccination. Age, medical insurance, and vaccine safety were significant determinants of COVID-19 VWR for both the paid and free vaccine. Income, occupation, and vaccine effectiveness were significant determinants of COVID-19 VWR for the free vaccine. Overall, the authors suggested targeting people over 58 years without medical insurance to improve COVID-19 VWR as well as the high-income groups for free COVID vaccine, and nationwide measures to educate people and increase awareness were needed. Overall, the article is well written and presents substantial data, the results are well explained, and a good discussion of the presented data and its comparison with previous studies from other parts of the world is provided. Additionally, the author also points out the strengths and weaknesses of the article and future steps to improve. Thus, I think the manuscript will be a valuable addition to COVID-19 vaccination research to understand VWR and measures to improve. I have no hesitation in accepting the manuscript for publication, once the below minor comments are attended to-
- Section 2.1 needs to be improved and explained a bit more in detail. The random selection of provinces and cities is a bit vague, and thus, needs to be explained a bit more – how was it picked? Also, it says 2700 adults were interviewed that yielded 2377 respondents, the change in the number of participants is not clear.
- When was this study conducted? For how long-duration of the study, and the order of the participants- which group/province, city, etc., were interviewed first?- All of these needs to be included.
- When was China's free vaccination policy introduced? And were these interviews conducted after the policy was issued? Or under the assumption that such policy might be issued?
- Why was the urban to rural ratio kept around 3:2? This should be explained.
- Figure 1: add a description to the y-axis
- Page 1, line 30, change "unwilling" to "unwillingness"
- Page 2, line 74-75- change –"identified the reasons participants vaccinated/not vaccinated" to "identified the reasons participants will get vaccinated/not vaccinated"
- Page 9, line 270- change "draw" to "drawn"
Author Response

(The authors gave the same response as above.)
